



# Effects of weather and climate on fluctuations of grain prices in southwestern Bohemia, 1725–1824 CE

Rudolf Brázdil[1,2], Jan Lhoták[3], Kateřina Chromá[2], Petr Dobrovolný[1,2]

[1]Institute of Geography, Masaryk University, Brno, Czech Republic
[2]Global Change Research Institute, Czech Academy of Sciences, Brno, Czech Republic
[3]Department of Historical Sciences, University of West Bohemia in Pilsen, Czech Republic

*Correspondence to*: Rudolf Brázdil (brazdil@sci.muni.cz)

**Abstract.** Grain prices in early modern Europe reflected the effects of weather on crop yields and a complex array of societal and socio-economic factors. This study presents a newly-developed series of grain prices for Sušice (southwestern Bohemia,
Czech Republic) for the period 1725–1824 CE, based on various archival sources. It aims to analyze their relationships with weather and climate, represented by temperature, precipitation, and drought (self-calibrated Palmer Drought Severity Index, scPDSI) reconstructions, as well as particular weather extremes and anomalies reported in documentary evidence. Wheat, rye, barley, and oats series in Sušice showed high mutual correlations. The highest prices in mean annual variations typically occurred from May to July before the harvest, while prices usually declined afterwards. Wheat, rye, and barley prices were
significantly negatively correlated with spring temperatures and positively correlated with scPDSI from winter to summer. This indicates that wetter winters, cooler and wetter springs, and wetter summers contributed to higher prices. The extremely high grain prices in the years 1746, 1771–1772, 1802–1806, and 1816–1817 were separately analyzed with respect to weather/climate patterns and other socio-economic and political factors. The results obtained were discussed in relation to data uncertainty, factors influencing grain prices, and the broader European context.

**1 Introduction**

The production of grain, a vital food source, significantly influenced society in early modern Europe (e.g., Hoskins, 1968; Persson, 1999). The availability of grain largely depended on its yields (e.g., Huhtamaa and Helama, 2017), a factor that, along with many others, was reflected in grain prices (e.g., Wrigley, 1989; Scott et al., 1998; Edvinsson, 2012). While the demand for grain was characterized by minimal elasticity and driven by population size, the market supply of grain varied
seasonally and interannually, showing a negative correlation between the quantity of grain produced and its prices (Esper et al., 2017). Weather conditions significantly influenced grain yields during all cereal growth stages, including sowing, growth, maturity, and harvest, as evidenced in available documentary sources (e.g., Šůla, 1969). The effects of climate change, which averages annual and longer-term weather variability, appear to be more complex (e.g., Holopainen et al., 2012; Huhtamaa et al., 2015; Esper et al., 2017; Skoglund, 2022). Ljungqvist et al. (2022) reported that studies analyzing the
influence of long-term climate variability on early modern grain yields and prices span a broad spectrum, ranging from





climate determinism to outright rejection of potential climate effects. More recently, Ljungqvist et al. (2023) highlighted the need for further studies on the relationship between grain quantity and quality, their links to climate, and the subsequent effects on grain prices in historical contexts.

The development of modern historical climatology, based on high-quality documentary evidence, climate reconstructions, and the close collaboration of climatologists and historians (e.g., Brázdil et al., 2005, 2010; White et al., 2018, 2023; Ljungqvist et al., 2021; Pfister and Wanner, 2021), has opened new perspectives to prove the robust influence of weather and climate on grain harvests and prices. This research direction, represented over the past decades by numerous studies from various countries, has demonstrated this relationship. For instance, studies conducted in Switzerland (Pfister, 1988), Germany (Bauernfeind and Woitek, 1999), Sweden (Edvinsson, 2009), England (Campbell, 2010; Campbell and Ó Gráda, 2011; Pribyl, 2017; Bekar, 2019), the Burgundian Low Countries (Camenisch, 2015), Finland (Huhtamaa, 2018), Sweden, Switzerland, and Spain (Ljungqvist et al., 2023), and across Europe (Esper et al., 2017; Ljungqvist et al., 2022), have all shown such relationships.

Novotný (1963) analyzed wheat and rye prices in Moravia (eastern Czech Republic) before 1618 CE, focusing on their dependence on "the natural character of the year". Kazimír (1968) attributed the increase in grain prices in the late 18th century to decreased production due to crop failures, but also suggested the influence of factors independent of yields in years with average harvests. Subsequent studies on grain prices in the Czech Republic primarily considered societal and socio-economic factors influencing their changes (e.g., Pšeničková, 1974; Borská-Urbánková, 1977, 1991; Křivka, 1977; Petráň, 1977; Tlapák, 1977; Hájek, 1981; Vorel, 1999). Exploring the potential impact of adverse weather on grain prices, Brázdil and Durďáková (2000) analyzed such relationships for grain series from three Moravian towns between the 16th and 18th centuries. Various grain price series in the Czech Republic were also employed to illustrate the human impacts of the volcanic eruptions of Lakagígar (Iceland) in 1783 and Tambora (Indonesia) in 1815 (Brázdil et al., 2017), as well as to demonstrate the long-term impacts of droughts on agriculture (Brázdil et al., 2019).

The aim of the current study is to analyze fluctuations in grain price series in Sušice, a town in southwestern Bohemia (Czech Republic), during the 1725–1824 CE period, in relation to potential effects of weather extremes and climate variability, while considering other influential societal and socio-economic factors. This interdisciplinary study seeks to fill existing research gaps in this part of Central Europe, as Czech grain price series were not included in pan-European studies such as those by Esper et al. (2017) or Ljungqvist et al. (2022).

## 2 Data

### 2.1 The Sušice region

The royal town of Sušice (Fig. 1), located at the foothills of the Šumava Mountains in southwestern Bohemia at an altitude of approximately 470 meters above sea level, historically belonged to the Prácheň Region, a part of the Czech Kingdom within the Austrian Habsburg monarchy (see Fig. 2 for places in the Czech Republic). In the 18th century, Sušice had a



population of around 2,500 and was the landlord of a smaller domain (Lhoták, 2018). Following a fire in 1707, the local economy of Sušice faced challenges. Its remote location, away from the network of imperial roads, hindered trade

development and resulted in local craftsmen producing mainly for the local market. According to the Teresian Land Registry, Sušice served as a market center for peasants from 91 localities. As the town was not situated between fertile and infertile areas, the local market experienced relatively less demand and lower frequency compared to most market centers in southern Bohemia (e.g., Petráň, 1977). Sušice organized four annual markets, but grain was sold at the weekly market every Monday, with an additional grain market added every Friday from 1785 (Lhoták, 2018). While in Western Europe, wheat

was considered the most important bread cereal (Hoskins, 1964; Campbell and Ó Gráda, 2011), in Central Europe, rye was more dominant, covering the largest areas and exhibiting the most responsiveness to fluctuations in production and supply. In the Sušice region, wheat and rye were the main bread grains, sown in autumn as winter crops, while barley and oats were sown in spring as spring crops. The timing of the harvest, typically in July–August, varied depending on natural and other circumstances of the given year.

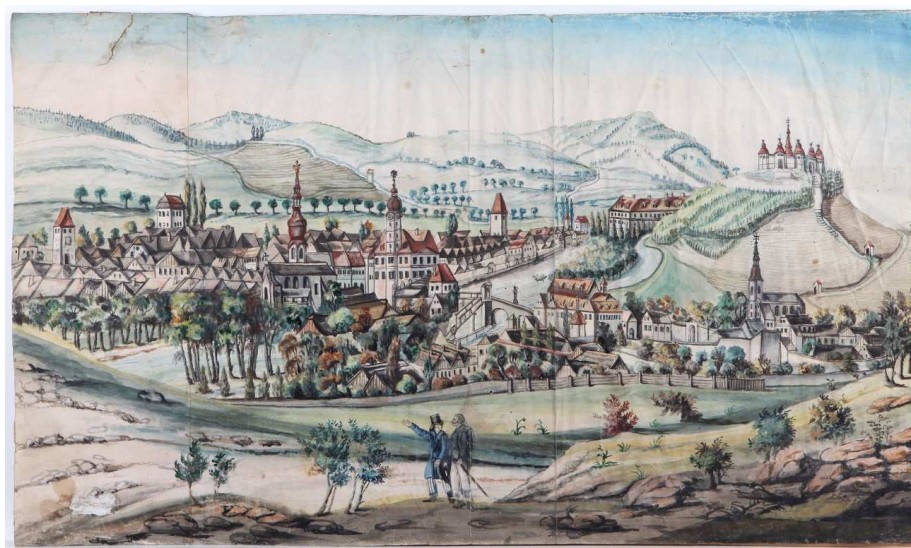


**Figure 1. The town of Sušice from the southeast – drawing by Augustin Maštovský from 1850, based on an earlier sketch by his father, Josef Maštovský, from 1832 (Muzeum Šumavy in Sušice).**



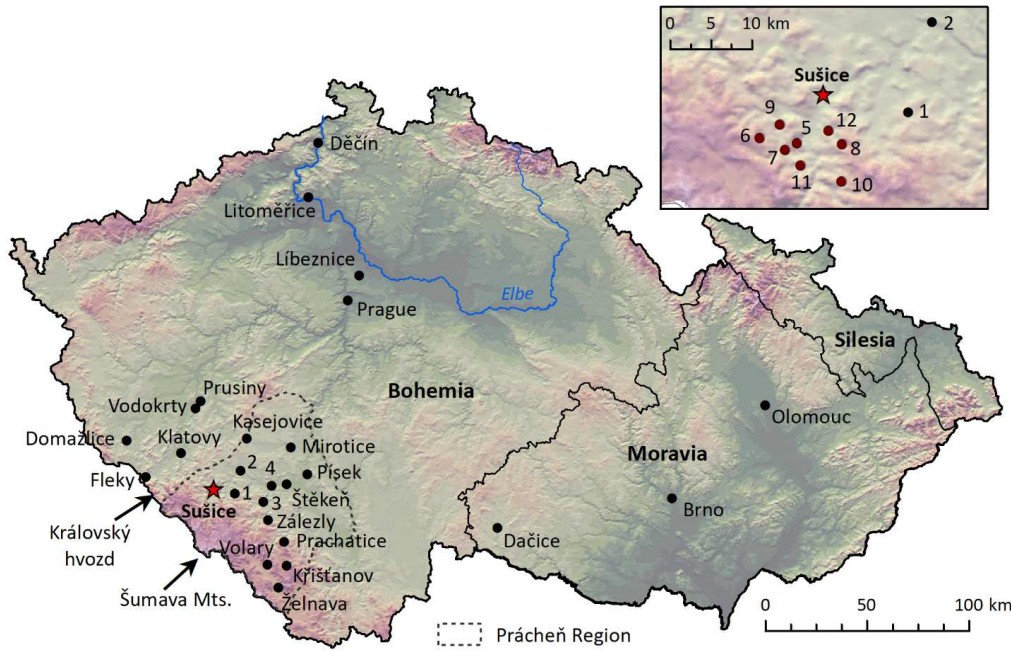

**Figure 2. Location of the town of Sušice and other places in the Czech Republic mentioned in the text. Key: 1 – Bukovník, 2 –**
**Horažďovice, 3 – Nihošovice, 4 – Strakonice, 5 – Dolejší Krušec, 6 – Dolejší Těšov, 7 – Hořejší Krušec, 8 – Humpolec, 9 – Jiřičná,**
**10 – Kašperské Hory, 11 – Palvínov, 12 – Platoř.**

The life in the town of Sušice and its surroundings during the 18th and 19th centuries was significantly influenced by three major wars:

(i) The War of the Austrian Succession (1740–1748)

This conflict began following the death of Emperor Charles VI, when Bavaria and Saxony rejected the "Pragmatic Sanction" and the succession of Maria Theresa to the Habsburg monarchy. King Frederick II of Prussia made territorial claims on Silesia. The subsequent involvement of France, England, and Spain in the conflict was driven not only by territorial interests but also by power struggles. The war ended with Maria Theresa's confirmation as empress, albeit at the cost of losing a significant part of Silesia to Prussia (e.g., Browning, 1993; Anderson, 1995).

(ii) The Seven Years' War (1756–1763)

This extensive military conflict pitted England, Prussia, Portugal, and some German states against France, the Holy Roman Empire (Austria and Saxony), Russia, Sweden, and Spain. Its initial goal was to redraw European borders. Although the Czech Lands permanently lost Kladsko (Kłodzko) and a large part of Silesia during this war, the borders of the main powers remained largely unchanged afterward (e.g., Hochedlinger, 2003; Szabo, 2007).



(iii) The Napoleonic Wars (1803–1815)

The Napoleonic Wars were a series of conflicts that followed the Wars of the Revolution in France (1789–1802). The principal adversaries were Great Britain, Russia, Prussia, and the Austrian Empire, with battles often occurring on or near their territories. The Czech Kingdom bore a significant burden in supporting the Austrian and, to some extent, the coalition armies. French troops also moved through Czech territory in 1805, 1809, and 1813 (e.g., Šedivý, 2001; Gates, 2003; Esdaile,
100   2019).

Concerning the climatic patterns of the Sušice region, it falls under the Köppen classification type Cfb – temperate broadleaf deciduous forest (Tolasz et al., 2007). Meteorological records for Sušice, dating back to 1961 (In-počasí, 2023), show a mean annual temperature of 7.4 °C and an annual precipitation of 658 mm. Monthly mean temperatures vary from –1.8 °C in January to 16.8 °C in July (Fig. 3a,b). The lowest mean monthly precipitation, at 32 mm, occurs in January and February,
while the highest is in June (88 mm), followed by August (86 mm) and July (80 mm). The mean onset of the harvest in the Sušice region from 1951 to 1980, as recorded by the Czech Hydrometeorological Institute's (CHMI) phenological stations in Mirotice (420 m asl) and Nihošovice (530 m asl) and derived from the FENODATA database, varied as follows: for winter rye between 26 and 29 July, for winter wheat between 2 and 8 August, for spring barley after 3 August (data solely from Mirotice), and for oats between 8 and 14 August. However, the latest harvests for barley and oats sometimes extended into
early September (Fig. 3c).

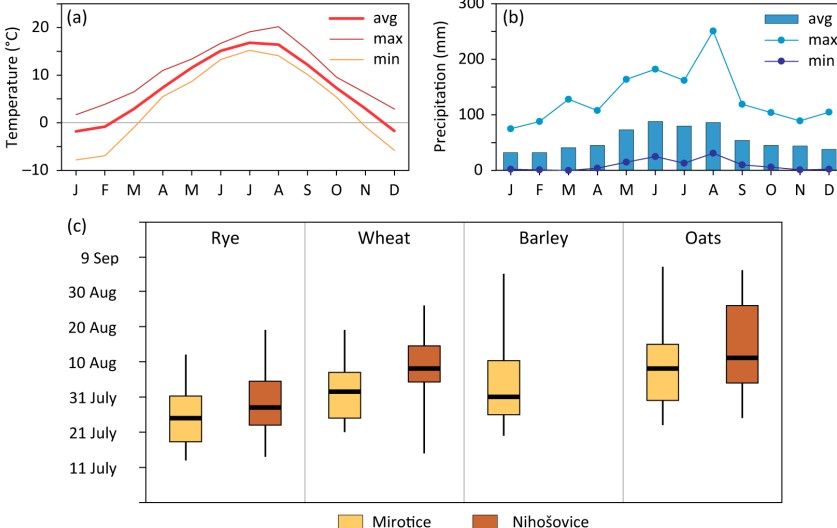

**Figure 3. Annual variations of monthly mean temperatures (a) and precipitation totals (b) in Sušice (avg – monthly mean, max – absolute monthly maximum, min – absolute monthly minimum) (data source: In-počasí, 2023); (c) box-plots (maximum and minimum, upper and lower quartiles, median) of the start dates of harvest for four main cereals in Mirotice and Nihošovice**
**(excluding barley) for the period 1951–1980 (data source: CHMI database FENODATA).**



## 2.2 Grain price data

The market dynamics in Sušice were notably affected by the irregular presence of the army in the town and its surroundings. Beginning in 1725 CE (Fig. 4), the army's needs necessitated the collection of data on maximum grain prices from across Bohemia. Reports from individual market places were relayed through regional offices to the Estates Land Committee

(*Stavovský zemský výbor*) in Prague and, during 1748–1763, to the Bohemian land government (*Gubernium*) at the same location. These records are now housed in the central collections (archive groups) of the National Archives (archival sources AS1, AS2). However, many market places maintained their own records (with some gaps) as Sušice from 1725–1789 (AS6). With few exceptions, these data are consistent with those in reports preserved for 1745–1748 and 1748–1763 (with gaps) in the central collections of the National Archives (AS1, AS2).

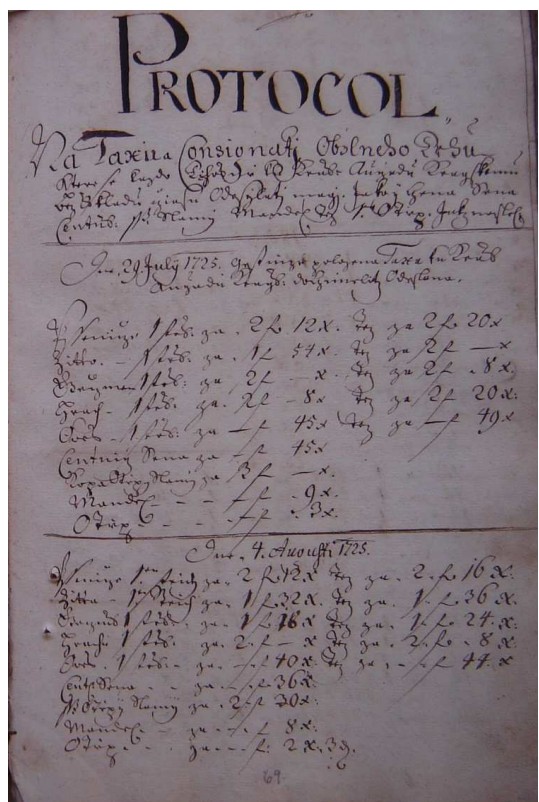


**Figure 4. An example of grain price records from July and August 1725 in Sušice (AS6, folio 69r).**



An additional source of price data is a summarized list of grain prices for the years 1789–1817, compiled for the Stable Land Registry (AS8; Tlapák, 1977) and published by Bisingerová et al. (1984). The records were transcribed onto pre-printed forms using the Lower-Austrian cubic measure (*měřice*), with prices listed in silver coins (until 31 October 1799), banknotes
– *bankocetle* (until 14 March 1811), and then Vienna currency banknotes (*Wiener Währung*). These records occasionally overlap with a collection of monthly reports from the Prácheň Regional Office for the period 1803–1827, also stored in the town archive in Sušice (AS7). There are occasional discrepancies between these records and the preceding source (AS8).

To compare and complement the grain price series in Sušice for the 1725–1824 period, series of wheat, rye, and barley prices (expressed in gulden) for the city of Prague were also collected and utilized (Schebek, 1873).

**2.3 Meteorological data**

Given that the period under analysis is only partially covered by instrumental observations from the Czech Lands, which began in 1775 in Prague-Klementinum for temperatures and in 1803 in Brno for precipitation (Brázdil et al., 2012), this study instead utilized reconstructed series that combine documentary data with these and other instrumental measurements. This includes the monthly temperature series for Central Europe (Dobrovolný et al., 2010), which is representative for
Bohemia (*cf.* Brázdil et al., 2022a), and the seasonal precipitation series for the Czech Lands (Dobrovolný et al., 2015). To assess the combined effects of temperature and precipitation, both reconstructions were employed to create series of four different drought indices (Brázdil et al., 2016a). For this study, the self-calibrated Palmer Drought Severity Index (scPDSI) was used. For information on weather extremes and anomalies in various localities in Bohemia, the study relied on documentary data from the historical-climatological database of the Institute of Geography, Faculty of Science, Masaryk
University in Brno.

**3 Methods**

**3.1 Compilation of grain price series**

The prices of the four main cereals were recorded in florens (gulden) and kreutzers (1 floren = 60 kreutzers) for the Bohemian *korec* (93.587 liters) until 1764 and the Lower-Austrian *měřice* (61.487 liters) from 1765 onwards. To account for
the difference in volume units, a conversion ratio of 1.522 to 1 was applied to recalculate prices from the Bohemian *korec* to the Lower-Austrian *měřice*.

Challenges in creating the price series arose due to the discrepancy between the nominal and real values of the currency (*disagio*), especially when metal coins, predominantly used for grain purchases for much of the 18th century, differed in value from the circulating *bankocetle*. From 1799 CE, and prior to the declaration of state bankruptcy in 1811 (Štaif, 2017),
the *disagio* was over 800–1200 % (for one gulden in silver, the equivalent was 8 guldens in *bankocetle*). Official conversions considered a smaller *disagio* of up to 500 %, and a 1 to 5 ratio was applied from April 1811 for exchanging *bankocetle* into



Vienna currency. However, the *disagio* of the new currency continued to rise relative to metal, stabilizing only in autumn 1818 at a value of 250 in relation to the metal currency.

After the aforementioned corrections to existing data, series of annual wheat, rye, barley, and oats prices for Sušice
between 1725 and 1824 CE were computed. This involved averaging all available corresponding prices for each year, collected from both central and local sources, with varying amounts of data in individual months. Due to the high correlations of monthly prices in Sušice with those in Horažďovice, Prachatice, and Strakonice during the 1745–1757 period, linear regression was used to supplement missing Sušice prices with additional data for some months in 1752 and 1753. This included wheat prices for 7 months in each year, rye for 5 and 7 months, barley for 5 months each, and oats for 6 and 7
months, respectively. Given the very high correlations between the Sušice and Prague price series (0.97 for wheat, 0.95 for rye, and 0.93 for barley), missing prices for these three cereals in 1742, 1773–1777, 1778 (only wheat and rye), 1784, and 1786–1788 were filled in using linear regression between these two locations. For the missing oats prices in 1742 and 1773–1777, their strong correlations with the other three cereals in Sušice were utilized to complete the data, averaging three calculated versions for each year.

### 3.2 Statistical analyses

Temporal fluctuations in the grain price series for Sušice and Czech Lands temperature, precipitation, and scPDSI series for 1725–1824 CE were visualized along with their linear trends calculated using the non-parametric Theil-Sen method, which offers robustness against outliers in time series (Sen, 1968; Theil, 1992). The non-parametric Mann-Kendall test was additionally used to assess the statistical significance of these linear trends (Mann, 1945; Kendall, 1975). All series were
detrended using a high-pass filter. Mean annual variations of the grain price series were computed for the July 1725–July 1738 period (complete monthly data are available, except for July 1737) and represented as box-plots (maximum and minimum, upper and lower quartiles, median) for individual months.

The study also analyzed the grain price series to identify years when their statistical properties underwent significant changes. The most probable years of changes in the mean, known as changepoints, were identified using the binary
segmentation algorithm (Scott and Knott, 1974) and the Bayesian Information Criterion, BIC (Bai and Perron, 2003), to prevent overfitting. Initially, a single changepoint test statistic was applied to the entire dataset. If a changepoint was detected, the time series was divided at that point, creating two new subsets for further analysis. This procedure was repeated on each subset. The method was employed as implemented in the changepoint R-package (Killick and Eckley, 2014).

Relationships between individual grain price series and between them and climate variables (seasonal temperature,
precipitation, and scPDSI) were evaluated using Pearson correlation coefficients, with statistical significance determined by the t-test ($p < 0.01$, $p < 0.05$ or $p < 0.10$). Climate variables were expressed as series of deviations from the 1961–1990 reference period, which was preferred over the more recent 1991–2020 period, already significantly influenced by global warming (see Brázdil et al., 2022b). Overall arithmetic mean and standard deviation of the high-pass filtered grain price series were used to select years with extremely high prices. These were years where prices exceeded the mean plus 0.5, 1.0,



1.5, and 2.0 times the standard deviation of each grain price series. For years with extremely high grain prices, corresponding values of the three climate variables were categorized in relation to multiples of standard deviations calculated from the 1961–1990 reference period.

Superposed Epoch Analysis (SEA) was utilized to assess the significance of mean climate conditions as potential triggers of extremely high grain prices. SEA is commonly employed to evaluate the mean response of climate variables to specific

events, such as volcanic eruptions (e.g., Esper et al., 2013), and was used by Esper et al. (2017) to investigate historical famines and the subsequent variability of grain prices in Europe. In our study, the years with extremely high grain prices in Sušice were designated as events, and SEA was applied to determine if specific seasonal climate conditions in the event years and the preceding years could have contributed to their occurrence.

Seasonal climate variables (temperature, precipitation, scPDSI) from the Czech Lands were first converted into z-scores.

Years in which grain prices exceeded the arithmetic mean plus one standard deviation were selected as extreme cases. The study then examined mean anomalies for these variables three years before and three years after the years with extremely high prices. The composite matrix created for this analysis included the years with extremely high prices in rows (event years) and the analyzed climate variables in columns (lag years). A random bootstrapping approach was employed to generate 500 unique versions of the composite matrices, each drawing unique subsets of event years randomly without

replacement (Rao et al., 2019). The density function and its percentiles were calculated for each column (i.e., for years from lag −3 to lag +3) and for each normalized composite matrix. A mean data anomaly for temperature, precipitation, and scPDSI exceeding the 95th percentile for each lag year was considered statistically significant (for further details, see Rao et al., 2019).

## 4 Results

### 4.1 Fluctuations in grain prices

The fluctuations in mean annual prices of wheat, rye, barley, and oats in Sušice during the 1725–1824 CE period are depicted in Fig. 5. These show highly consistent inter-annual fluctuations among all four grain price series, as indicated by very high and statistically significant ($p < 0.01$) Pearson correlation coefficients. The correlations were particularly strong between the three main crops (0.98 for rye and barley, 0.97 for wheat and rye, and 0.95 for wheat and barley), while

correlations with oats were slightly lower (0.88 with wheat and rye, and 0.89 with barley). Wheat consistently had the highest prices, followed by rye and barley, with oats having the lowest prices, as shown by the corresponding mean values (Table 1). The arithmetic mean of prices exceeded their median for all four cereals. The absolute maximum prices for all cereals were uniformly recorded in 1805. The absolute minimum prices occurred between 1733 and 1735 (1733 for oats, 1734 for rye, and 1735 for wheat and barley). Rye prices showed the highest variability, followed by barley, wheat, and oats,

according to the coefficient of variation. All cereal price series experienced statistically significant ($p < 0.01$) increasing linear trends, indicative of long-term inflation. After detrending the four grain price series using a high-pass filter, the most





notable price peaks occurred in 1805, 1817, 1771, and 1746 (Fig. 5b). Statistically significant changes in the mean of the detrended series for the 18th century revealed two levels of grain price series with a changepoint in 1773 for wheat, rye, and barley, and four levels for oats with changepoints in 1737, 1747, and 1756. The significant variability in grain prices during

the first quarter of the 19th century was also evident, with six different levels for the three reported cereals (changepoints in 1801, 1804, 1806, 1815, 1817, and 1818) and four levels for oats (changepoints in 1801, 1806, 1815, and 1817).

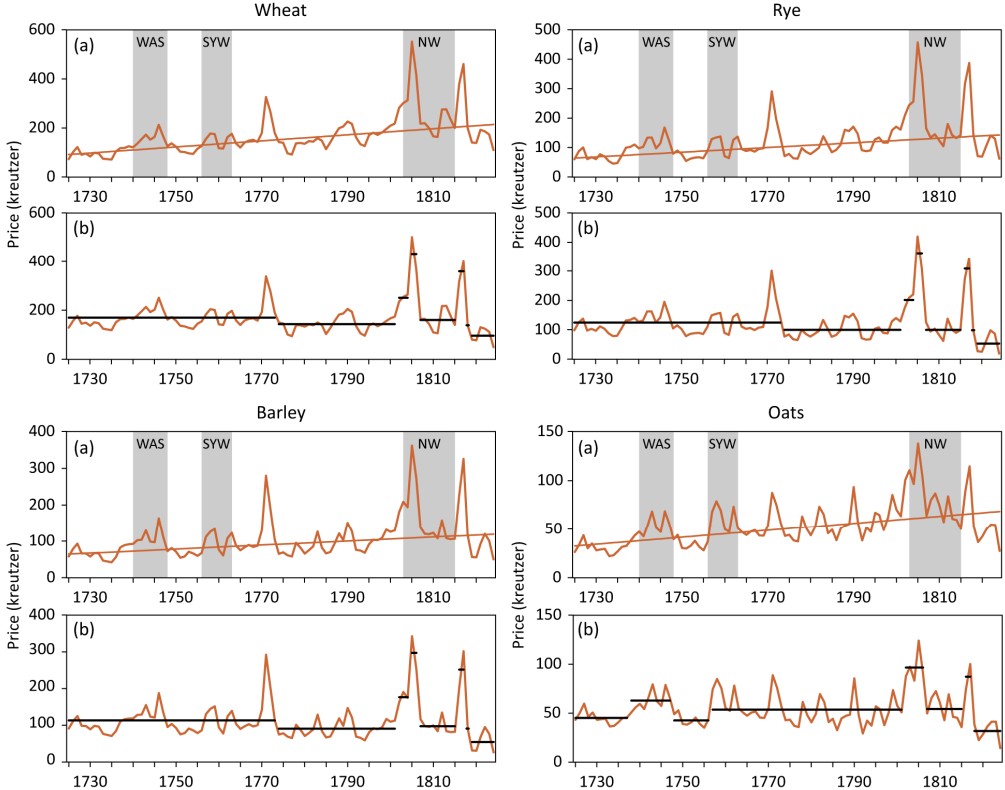

**Figure 5. Fluctuations in mean annual prices of wheat, rye, barley and oats (kreutzers per Lower-Austrian *měřice*) in Sušice during the period 1725–1824 CE: (a) compiled series, including linear trends (refer to Table 1 for trend values) and grey bands**
**indicating war years: WAS – the War of the Austrian Succession, SYW – the Seven Years' War, NW – the Napoleonic Wars; (b) detrended (high-pass filter) compiled series. The solid black line in part (b) represents the variation in statistically significant changepoints in the mean of each series.**





**Table 1. Statistical characteristics of grain-price series in Sušice for the 1725–1824 CE period. Key: SD – standard deviation, CV – coefficient of variation. Grain prices are expressed in kreutzers per Lower-Austrian *měřice*. Statistical significance of linear trends (expressed in kreutzers per 10 years): \*\*\* p < 0.01.**

| Cereal | Mean | Median | Maximum | | Minimum | | SD | CV (%) | Linear |
|--------|------|--------|---------|------|---------|------|------|--------|--------|
| | | | value | year | value | year | | | trend |
| wheat | 169.3 | 150.0 | 551.9 | 1805 | 71.2 | 1735 | 79.8 | 47.1 | 12.4\*\*\* |
| rye | 122.3 | 102.5 | 456.9 | 1805 | 45.6 | 1734 | 69.5 | 56.8 | 8.0\*\*\* |
| barley | 105.0 | 92.1 | 362.1 | 1805 | 41.8 | 1735 | 55.7 | 53.1 | 5.5\*\*\* |
| oats | 54.4 | 50.2 | 137.7 | 1805 | 22.1 | 1733 | 21.9 | 40.2 | 3.6\*\*\* |

The mean annual variations in the monthly prices of wheat, rye, barley, and oats in Sušice during the 13-year period from July 1725 to July 1738 (excluding July 1737) are depicted in Fig. 6 using box-plots. Generally, the prices for all grains were

highest from May to July, just before harvest, and then declined, with the lowest values observed in August for barley, September for wheat, and September to October for rye and oats. Following the months with the lowest prices, there was a slight increase in values up to December, after which the prices remained relatively stable throughout the winter and spring months, until May. The months with the lowest prices also exhibited the lower variability in prices, including July.

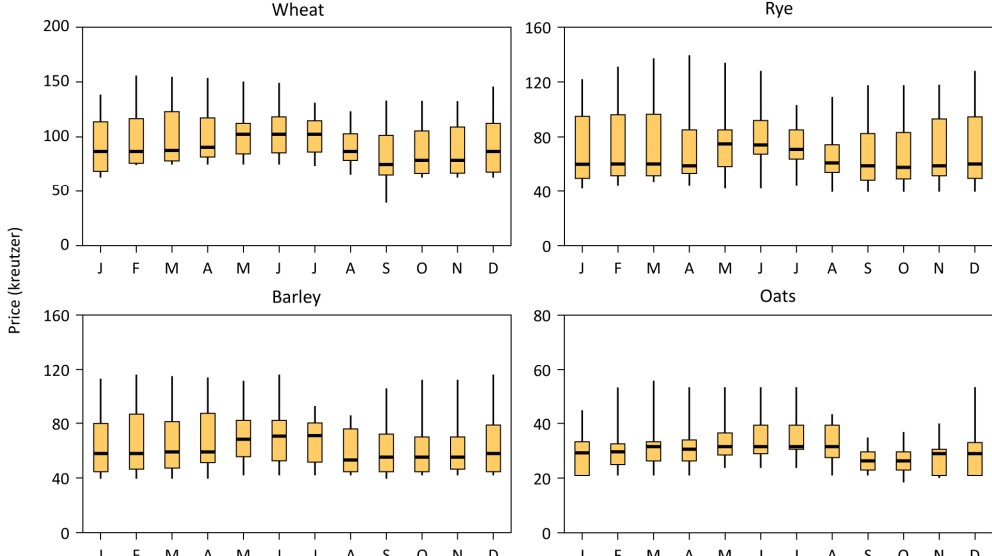

**Figure 6. Mean annual variations of wheat, rye, barley, and oats prices (kreutzers per Lower-Austrian *měřice*) in Sušice, July 1725–July 1738. Displayed as box-plots showing maximum and minimum, upper and lower quartiles, and median for individual months.**





### 4.2 Climate, weather, and grain prices

### 4.2.1 Long-term climate fluctuations

To assess climate variability and its potential impacts on grain prices, Fig. 7 presents the fluctuations in seasonal winter (DJF), spring (MAM), and summer (JJA) temperatures, precipitation totals, and droughts (as indicated by scPDSI) for the Czech Lands during the 1725–1824 CE period. These fluctuations are expressed as deviations relative to the 1961–1990 reference period. The DJF temperatures showed the highest interannual variability among the three seasons, with notable positive deviations in the 1790s. DJF precipitation deviations, in contrast, exhibited slight fluctuations around the 30-year

mean (by approximately ±50 mm). MAM temperatures were significantly below the reference period around 1740 and 1785, whereas MAM precipitation fluctuations were similar to DJF totals, with more frequent deviations below −50 mm. JJA temperatures had higher values in the second half of the 1720s (accompanied by below-mean precipitation) and negative deviations (with higher precipitation totals) in the 1810s. Notable dry periods in all three seasons occurred during the second half of the 1720s, the second half of the 1770s to early 1780s, and in the 1790s. The climate series under study exhibited

very small or even zero linear trends over this 100-year period, all statistically non-significant.

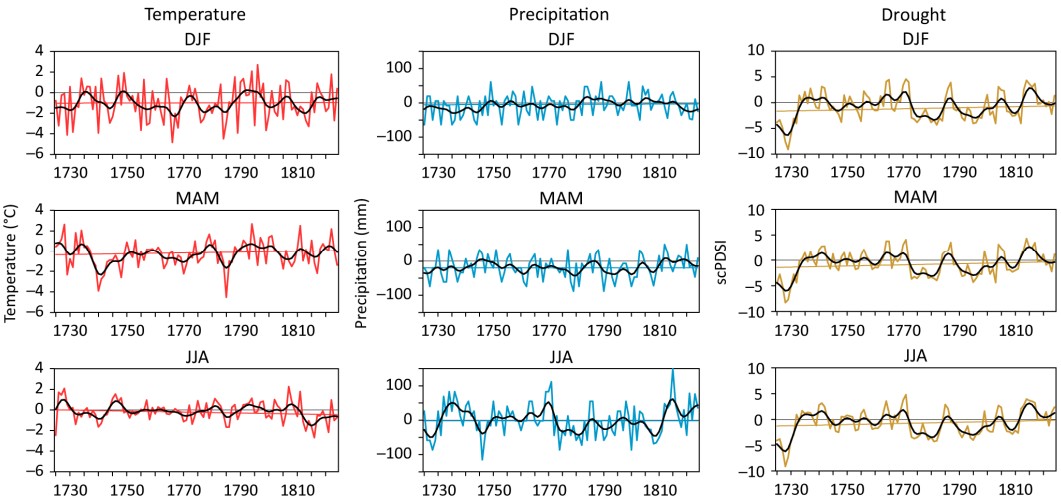

**Figure 7. Fluctuations and linear trends in mean winter (DJF), spring (MAM), and summer (JJA) temperatures, precipitation totals and scPDSI in the Czech Lands during the period 1725–1824 CE. Displayed as deviations relative to the 1961–1990 reference period and smoothed with a 10-year Gaussian filter.**

The correlation analysis between the detrended series of grain prices in Sušice and Czech DJF and JJA temperatures during 1725–1824 revealed no significant relationships, as indicated by very low and non-significant Pearson correlation coefficients, which fluctuated slightly around zero (Table A1). However, a stronger signal was observed when comparing grain prices with MAM temperatures, showing statistically significant negative correlations: −0.24 (p < 0.05) for barley and





oats, –0.25 (p < 0.05) for wheat, and –0.26 (p < 0.01) for rye. This suggests that higher prices correlated with cooler MAM
temperatures and vice versa. Similar weak and non-significant correlations, as seen with DJF and JJA temperatures, were
also found for seasonal precipitation totals. The highest correlation was only –0.14 for oats in JJA. However, the combined
effect of both temperature and precipitation on grain prices was evident in the higher statistically significant correlation
coefficients with scPDSI. The correlations decreased from DJF (rye 0.35, wheat 0.34, and barley 0.31, p < 0.01) to MAM
(0.31, 0.31, and 0.27, respectively, p < 0.01) and JJA (0.29 and 0.28, both p < 0.01, and 0.25, p < 0.05, respectively). This
indicates that higher prices were associated with wetter conditions and lower prices with drier weather patterns. For oats, the
only statistically significant correlation was with DJF scPDSI (0.17, p < 0.10).

However, good or bad harvest of the given year is reflected also in grain prices of the months in the following year until new
harvest. To evaluate such potential effects, series of climate variables were correlated with prices shifted by one year into the
past (i.e., always with prices in subsequent years) (Table A2). Significant correlations of grain prices with climate of
preceding year were calculated for MAM temperatures (rye and barley), DJF and JJA precipitation (except oats for JJA), and
all seasonal scPDSI (all cereals). Except MAM temperatures, related significant correlation coefficients were higher than in
correlating series without time shift (see Table A1).

### 4.2.2 Extremely high prices

Thresholds were established for identifying years with extremely high grain prices in Sušice, using means in detrended grain
price series increased by 0.5, 1.0, 1.5, and 2.0 multiples of the related standard deviation (Table A3). The focus was on years
where prices exceeded 1.0 standard deviation for further analysis. The selected years – 1746, 1771–1772, 1802–1806, and
1816–1817 – align well with the highest detrended prices illustrated in Fig. 5b. These years were further analyzed in relation
to DJF, MAM, and JJA climate variables, also considering the year preceding these selected years (Table 2). Climate
variables, characterized by deviations from the corresponding 1961–1990 means and compared to multiples of standard
deviations, were supplemented, in some instances, by documentary reports from Sušice and other nearby and distant
locations in Bohemia to detail related weather patterns.

**Table 2. Mean winter (DJF), spring (MAM), and summer (JJA) climate variables (T – temperature, P – precipitation, D – self-calibrated Palmer Drought Severity Index) for the years with extremely high grain prices (in bold) and the preceding year in Sušice during the 1725–1824 CE period. Climate variables are expressed as deviations with respect to the 1961–1990 period and**
**evaluated compared to multiples of corresponding standard deviations (SD) in the same periods: * >1SD, ** >2SD, *** >3SD.**

| Year | DJF | | | MAM | | | JJA | | |
|---|---|---|---|---|---|---|---|---|---|
| | T | P | D | T | P | D | T | P | D |
| **1746** | | | | | | | | | |
| 1745 | –2.4* | 5.7 | –0.7 | –0.9 | 33.7* | 1.2 | 1.1* | –1.8 | –0.0 |
| **1746** | –1.0 | –36.1* | –1.5 | –0.4 | 20.2 | –1.3 | 1.5** | –115.1** | –2.3* |
| **1771–1772** | | | | | | | | | |





| 1770 | −1.0 | 19.6 | 3.4* | −2.2** | −20.4 | 3.1** | −1.1* | 83.2* | 3.8** |
|------|------|------|------|--------|-------|-------|-------|-------|-------|
| **1771** | −0.6 | 5.7 | 4.6** | −1.3* | 6.6 | 4.1** | −1.0* | 111.5** | 4.8*** |
| **1772** | 0.7 | 5.7 | 4.0** | −0.2 | −20.4 | −0.1 | 0.5 | −44.3 | −0.6 |
| **1802–1806** | | | | | | | | | |
| 1801 | 0.2 | −8.2 | −2.4* | 1.5* | −47.4** | −2.7* | −0.9* | 40.7 | −2.4* |
| **1802** | −1.8* | −8.2 | −1.7* | 0.0 | −47.4** | −1.7* | 1.0* | −58.5* | −2.4* |
| **1803** | −2.3* | −50.0* | −4.0** | −0.1 | 32.0* | −4.4** | 0.4 | 25.8 | 0.5 |
| **1804** | 0.7 | 48.5* | 3.1* | −0.3 | 19.8 | 3.2** | 0.1 | 18.1 | 2.7* |
| **1805** | −2.6* | 23.2 | 2.5* | −1.8* | −10.9 | 2.4* | −1.6** | −16.0 | 2.3* |
| **1806** | 1.2 | 38.7* | 2.3* | 0.1 | −16.8 | 1.8* | −0.4 | −25.9 | −0.9 |
| **1816–1817** | | | | | | | | | |
| 1815 | 0.0 | 32.1* | 3.7** | 1.1* | −30.7* | 2.6* | −1.6** | 149.2*** | 3.7** |
| **1816** | −1.9* | 3.9 | 3.0* | −1.2* | −6.2 | 2.5* | −2.7*** | 45.3* | 3.5** |
| **1817** | 0.9 | −5.6 | 3.7** | −2.2** | 15.8 | 3.5** | −0.6 | 9.8 | 3.2** |

### The year 1746

Following the start of the War of the Austrian Succession in 1740, grain prices began to rise, peaking in 1746, and were also high in 1742–1743 and 1747 (Fig. 5, Table A3). According to grain price reports from Bohemia for 1745–1749 (Borská-Urbánková, 1977), the June 1746 prices for wheat, rye, and barley in Sušice reached the highest values in the entire country. Climatic data for the Czech Lands in 1746 showed a cooler and drier DJF and a slightly cooler and wetter MAM, followed by a warm and extremely dry JJA (Table 2). Numerous sources in Bohemia reported a severe drought, especially in June and July. For instance, František Václav Felíř recorded in Prague only two light rains during these two months and a rain-seeking procession held on 9 July (Vogeltanz and Ohlídal, 2011). Reports from July in Líbeznice mentioned dried-up springs and wells due to the dry and hot weather (Třebízský, 1885). The drought led to grain shriveling and frequent forest fires, as seen in southern Bohemia near Želnava, Volary, and Křišťanov (Berger, 1880/81). This harvest failure was reflected in the high grain prices, as noted in a manual from Kasejovice (Siblík, 1903) and other sources from the Czech Lands (see Brázdil and Trnka, 2015). According to records from Litoměřice, many watercourses dried up or had such low water levels that water mills ceased operation, causing a scarcity of flour and bread (Katzerowsky, 1887). The extremely low water level of the Elbe River at Děčín in 1746 was marked by a corresponding sign on the "Hunger Stone," alongside other extremely dry years (see Fig. 6 in Brázdil et al., 2019).

### The years 1771–1772

These years belong to the so-called "hungry years 1770–1772" in the Czech Lands (Brázdil et al., 2001). Very wet patterns expressed in scPDSI prevailed from DJF 1770 until DJF 1772 (particularly wet was in the year 1771), being accompanied mainly by colder MAMs and JJAs (Table 2). It caused two subsequent years of grain failure in 1770 and 1771. Soaked roads





and wet fields complicated sowing of winter crops already in the autumn 1769 (Třebízský, 1885). In 1770, on 19 March fell a great amount of snow which was lying for four weeks that all grain already sown extinct (e.g., Jílek, 1905; Řehák, 1912). But after this period followed drought continuing for seven weeks, i.e., sooner too wet and then too dry conditions led to very bad harvest (Skopec, 1907). The chronicle of Šebesta family from Klatovy reported for 1771 a crop failure after

frequent rains from May until July and a hunger continuing for the second year, lack of grain for autumn sowing, use of barley and oats for bred instead of rye, little beer and no spirits due to lack of grain, etc. (Hostaš, 1895). Similarly, a report from Sušice in 1771 mentioned crop failure in "*all countries*" and that "*people cried of great hunger*" (Lhoták, 2016, p. 492). Although reports from Klatovy mentioned before the 1772 harvest "*a great force of mice that devoured the second part of grain*", prospects for the new harvest were evaluated as good that "*a terrible hunger … will be averted by the Almighty*

*through great abundance of everything in fields and* [good] *yield*" (Hostaš, 1895, p. 282). But two years of grain failure combined with socio-economic situation in Bohemia were reflected in several human impacts, responses and consequences (see Brázdil et al., 2001; Pfister and Brázdil, 2006 for more details) among which significantly increased mortality (particularly in 1772) influenced natural development of population in the Czech Lands in such way, that pre-crisis level of population from 1770 was achieved as far as after 13 years (e.g., Fialová et al., 1996; Steinbachová, 2001). However, a deep

crisis of 1770–1772 had also a broader European scale (see Collet, 2019).

**The years 1802–1806**

These years of high grain prices, coinciding with the onset of the Napoleonic Wars in Europe in 1803, peaked sharply in 1805 (Fig. 5). ScPDSI reconstructions for the Czech Lands (see Fig. 7) show dry patterns from DJF 1801 to MAM 1803, which were then replaced by wet conditions lasting until MAM 1806. The year 1805 was also notably cooler (Table 2).

Georg Mayer, a farmer from Fleky, reported that around one ell of snow (approximately 78 cm) fell on 16 May 1802, remaining for several days, damaging cereals and trees. Mice also caused damage to grain that year and the previous one (Blau, 1908). Similar heavy snowfall was reported in places like Prusiny (AS3) and Zálezly (AS10), with annalists noting the destruction of winter rye, which had to be ploughed under. The Augustinian memorial book from Domažlice describes the cold DJF 1802/03 and high prices (Řehák, 1912, p. 133): "*The winter of 1803 was marked by extreme cold, causing*

*brooks to freeze to their bottoms and mills to cease operations. This led to a significant increase in flour prices, prompting bakers to progressively reduce the size of their goods. Additionally, the unpopularity of bankocetle, as people preferred coins, contributed to the high prices.*" František Tomáš Spillar (AS3), a teacher in Prusiny, added that in 1803, "*due to drought there was again no good harvest and also much* [grain] *was exported to Bavaria*". Following the wet winter of 1803/04, "*grain in many places withered and much rye had to be ploughed under during the spring*" (ibid.). In response to

the poor harvest of 1804, the Sušice town council purchased a large quantity of grain from Bavaria (Holík, 1885). In 1805, Domažlice also imported Bavarian grain to prevent starvation, with grain prices escalating higher than in 1804 due to poor harvests and war (Řehák, 1912). The chronicle from Zálezly (AS10) described the winter of 1804/05 as harsh and prolonged, with snow and cold around 5 May 1805 and a significantly delayed harvest, sometimes until 10 October. Also at Prusiny (AS3), the harvest was delayed; "*after St. Bartholomew* [24 August], *enough rye was still standing in the fields, unripe due*



*to rains and yet to be harvested,*" leading to widespread hunger and poverty as "*such high prices were never before seen*".
The year 1805 in Fleky was characterized as wet, cold, and detrimental to grain (Blau, 1908). Although 1806 saw a good
grain yield, unstable weather during harvest led to difficulties in storage, causing grain to rot or become musty in barns and
granaries (AS3). As Spillar noted, "*much such bad grain remained with rich and wealthy people who waited for high prices
as in preceding years*" (ibid.).

**The years 1816–1817**

In 1815, the Napoleonic wars in Europe ended, and the Tambora volcano erupted in April in the Lesser Sunda Islands,
Indonesia. This led to the infamous year 1816, known as "without a summer" (e.g., Luterbacher and Pfister, 2015). In 1815,
after a drier MAM, a cold and extremely wet JJA followed, while cold MAM and JJA also occurred in the subsequent two
years. The corresponding scPDSI values indicated extremely wet patterns in all three years (Table 2). Detailed descriptions

of the 1815–1817 years in the Czech Lands based on instrumental and documentary data can be found in Brázdil et al.
(2016b, 2017), supplemented by reports from the analyzed area. For example, Georg Mayer in Fleky reported a cold and wet
year in 1815 with a bad grain harvest and many mice (Blau, 1908). The next year, 1816, was also cold and wet, with frosts
significantly damaging blossoming cereals, and prices of new grain constantly rising (ibid.). Václav Jan Mašek from
Vodokrty reported rains from 8 June 1816 for eight weeks when only rarely any day was without rain (Urban, 1999). All

cereals became wet, yields were bad, prices were high, and hunger prevailed among the people. Due to a lack of grain, some
fields remained unsown in the autumn (ibid.). In 1816, grain became a subject of dispute between the Sušice town and its
serfs over the payment of feudal rents in agricultural products (mainly in grain); the peasants refused it due to its high price
(Lhoták, 2016). The Bukovník (AS9) chronicle noted a mild DJF 1816/17, when snow and frosts with changeable weather
only came from the beginning of April, complicating spring sowing and endangering winter cereals. It specifically reported

"*such numbers of beggars, coming every day as is impossible to describe*" (ibid.). Georg Mayer described the year 1817 in
Fleky as "*the year of poverty*" (Blau, 1908). In Sušice, the high grain prices continued until a good harvest in 1817 (Lhoták,
2016).

**4.2.3 Superposed epoch analysis of extremely high grain prices**

In the superposed epoch analysis (SEA) of extremely high grain prices in Sušice, temperature, precipitation, and scPDSI

anomalies (with 1961–1990 as the reference period) in the Czech Lands were examined. This analysis was based on years
when the mean of detrended series of wheat, rye, and barley prices increased by one standard deviation (see Table 2). Since
the price series for these grains, including the occurrence of extreme years, are very similar, the SEA revealed comparable
climate conditions preceding these extreme years (Fig. 8a). It was generally observed that years with extremely high prices
were often preceded by colder years in all the seasons studied. However, most of these negative temperature anomalies were

not statistically significant. Regarding precipitation, excluding the impact three years before the price increase, there was a
trend of higher precipitation totals. This increase was statistically significant ($p < 0.05$) for JJA one and two years prior to the
price increase. Similar to the findings in the correlation analysis (Sect. 4.2.1, Tables A1 and A2), the SEA showed a clear



influence of wetter conditions, as indicated by positive scPDSI anomalies, on the rise in grain prices. These positive scPDSI

anomalies were prevalent in all seasons. For DJF and MAM, these anomalies were statistically significant both in the year of

the price increase and the year before. In contrast, for JJA, the significant anomalies occurred one and two years prior to the

price hike. The situation differed slightly with extremely high oats prices (Fig. 8b). The climate characteristics of the three

years preceding these extreme price years showed that while DJF and JJA precipitation one year before the extreme prices

were significantly higher, scPDSI values significantly increased in the extreme year and the year before during DJF, one year

before the extreme in MAM, and one and two years before the extreme in JJA.

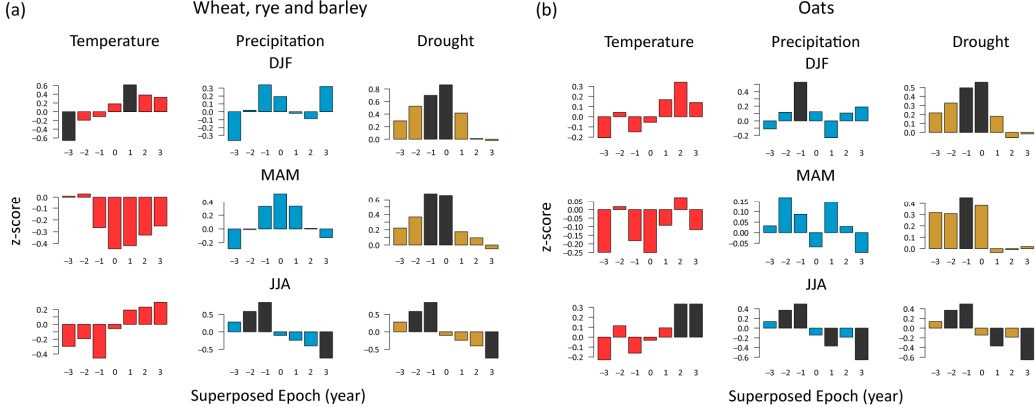

**Figure 8. Superposed epoch analysis of seasonal (DJF – winter, MAM – spring, JJA – summer) temperatures, precipitation, and scPDSI in the Czech Lands from 1725–1824 CE, aligned with years of extremely high prices (0 – event year) for wheat, rye, and barley (a) and for oats (b). The individual columns display mean climate conditions three years before and after the event years. Climate variables were transformed into z-scores, and significant anomalies ($p < 0.05$) are indicated in black.**


## 5 Discussion

### 5.1 Data uncertainty

Documentary data may suffer from temporal or spatial inhomogeneity, covering different time intervals and locations.

Sometimes the required data were not recorded at all, or related sources have been lost or destroyed over the years (e.g.,

Brázdil et al., 2005). This inhomogeneity complicates the creation of long-term series for certain variables, introducing

uncertainties, which also applies to grain prices. For the Sušice grain-price series, this issue manifested in the varying

number of records from which monthly means were calculated and subsequently used to compute annual means.

Additionally, in some years, no grain prices were available, and annual values had to be supplemented from the Prague

series, which showed very high and statistically significant correlations with Sušice (see Sect. 3.1). The bias incorporated

into the compiled Sušice series for these reasons may similarly affect the study of their relationships with climatic variables,

but it did not influence the selection of years with extremely high prices analyzed separately in Sect. 4.2.2 and 4.2.3.



When comparing grain prices with reconstructed series of three climate variables for the Czech Lands, it is also necessary to mention their uncertainties in relation to our study. While high spatial correlations of temperatures in Central Europe confirm the validity of using the temperature series (Dobrovolný et al., 2010) for southwestern Bohemia, the situation with precipitation is more complex. The seasonal precipitation series (Dobrovolný et al., 2015) was based on precipitation indices

created from documentary sources from various parts of the Czech Lands where such records were available. This means the related reconstruction represents average patterns for the entire country, not reflecting potential regional differences due to weak spatial correlations of precipitation totals (*cf*. Brázdil et al., 2021). Consequently, using both temperature and precipitation series for calculating scPDSI (Brázdil et al., 2016a) may also partly suppress regional scPDSI differences across the country.

### 5.2 Natural and socio-economic factors of grain prices

The success or failure of grain harvests directly impacted grain prices and subsequently could lead to food shortages and high prices of other goods. Harvest and crop yields in a particular year may have been influenced by disastrous weather events or anomalies (e.g., hail, flood, frost, drought, rain spells). For example, a hailstorm on 22 June 1760 caused significant damage to fields in Sušice and 16 other villages, and on 27 and 28 June, six more places were affected (AS5).

Similarly, on 14 June 1796, hail and water (from torrential rain) damaged fields of Sušice inhabitants as well as those in several surrounding villages (AS8). The harvest could have been complicated by the occurrence of rain spells. For instance, on 3 August 1737 in Domažlice, a large beseeching procession was organized from the town to St. Lawrence Chapel to pray for favorable weather because long rain spells impeded the drying of crops for harvesting (Řehák, 1912). Droughts, as the opposite extreme to rain spells, had other negative consequences for grain, influencing not only the grain itself but also

causing shortages of straw. For example, on 23 June 1758, the burghers of Klatovy intended to organize a beseeching procession for rain due to "*such a dry year, that grain could not ripen and a greater part withered than ripened*". But instead of rain, the town caught fire (Hostaš, 1895, p. 273). In 1782 in Prusiny, spring grain grew poorly due to frosts and winds, which were followed by drought lasting until 22 September. This caused a very poor harvest of barley and oats, which had to be pulled out with roots by hand instead of being cut (AS3).

Diseases and pests also significantly affected grain yields. For instance, in 1742, an infestation of mice was so severe that, as reported by the Šebesta family chronicle from Klatovy, "*three parts of the grain and straw in the fields were devoured, raising grain prices to six guldens. The mice reappeared in the autumn, leaving hardly a third of the grain. Consequently, people had to plough under winter cereals in spring* [1743] *and begin sowing spring crops*" (Hostaš, 1895, p. 273). A similar situation occurred in 1773 in Štěkeň, where peasant Antonín Pašek reported an abundance of mice damaging grain both

before the harvest and after autumn sowing, to the extent that "*fields were full of holes, and the grain was so devoured* [by mice] *that in many places, not a single small grain remained*" (Jílek, 1905, p. 95).

The natural conditions significantly influenced the resulting prices. The Sušice market circle included localities that, in the first half of the 18th century, were considered some of the most impoverished in Bohemia (Chalupa, 1969). While some





villages, such as Dolejší Těšov, Dolejší Krušec, Hořejší Krušec, Jiřičná, and Palvínov, did not sell their grain and used it
only for their consumption, soil infertility in higher altitudes forced people to buy grain for their sustenance. This was either
partial, as in the Kašperské Hory domain, or complete, with survival mainly through stockbreeding in localities like
Královský hvozd. For example, the servile villages of Sušice, Humpolec and Platoř, confessed in 1825 to only sowing spring
rye and oats (Lhoták, 2018).

The natural conditions and phenomena described above interacted with various societal and socio-economic factors that
influenced grain prices. Petráň (1977) identified key factors including wars, administrative decrees, corn reserves,
anticipated grain yields, the movement of grain into and out of the country, frequency of grain markets, and speculation.
Borská-Urbánková (1977) acknowledged the impact of good or bad harvests on grain prices, but only in years without war
conflicts. The presence of an army in a region was likely one of the most significant factors affecting grain prices. In Sušice,
winter quarters for soldiers were recorded in 1749 (four infantry companies), continuing in the early 1750s (1750/51,
1751/52, 1752/53, and 1754) and the 1760s (1763, 1765/66). Although the war conflicts described in Sect. 2.1 led to short-
term increases in grain prices (refer to Fig. 5), they did not have as clear negative effects as the significant decreases in
building activity in the Czech Lands (for comparison, see Fig. 3 in Kolář et al., 2022).

Wartime, with its increased grain demand, could lead to short-term developments in local grain commerce, often with
speculative intentions (Lhoták, 2022). Thürheim's infantry regiment was stationed in the town permanently from 1766 to
1779 (Wrede, 1898), significantly influencing local market dynamics. For instance, in the summer of 1771, the regiment's
commander requested permission for a second weekly market due to high prices, which he attributed to insufficient public
oversight and rampant trafficking. The town council countered by citing generally poor conditions in Bohemia, arguing that
peasants would not bring grain to the town without expectations of higher prices and would instead sell directly to coachmen
visiting villages. All grain brought to the town was immediately purchased by soldiers (AS4).

The aforementioned stays of domestic military forces were occasionally supplemented by short-term presences of enemy
armies, such as French soldiers in the Prácheň Region during 1742 and in the Klatovy Region during 1805 and 1809. Market
dynamics were further influenced by the location of main military depots. Between 1792 and 1815, such depots were
situated in Klatovy, Strakonice, and Písek, receiving grain from the Sušice region at officially established prices (Lhoták,
2022).

Demand for wheat and rye, key bread grains, was affected by the poor storability of flour. This limited the potential for
speculative buying, generally leading to stable demand. An increase in population heightened the demand for grain (Honc,
1977). While wheat was typically the most expensive grain, local demand could sometimes shift this trend. For instance, in
northern Bohemia during August and September 1757, the army's need for oats drove up their prices (Borská-Urbánková,
1977). In Sušice, oats also reached very high prices in 1757 and three additional years (1758, 1759, and 1762) during the
Seven Years' War (1756–1763), whereas for the other three cereals, the peak years were 1758, 1759, and 1763 (*cf*. Table
A3). Conversely, barley prices rose due to increased production in nobility-owned breweries, among other factors. Price



variations across regions are evident from government reports (*Gubernium*) in 1754–1755: the lowest prices for wheat were in central Bohemia, for rye in southern Bohemia, for barley in eastern Bohemia, and for oats in southwestern Bohemia.

Differences in grain prices between the south or southwest and northwest parts of Bohemia were also influenced by trade
relationships with border areas. In southwestern Bohemia, such contacts were minimal. The Teresian Land Registry from the first half of the 18th century indicated that, of 692 localities in the Prácheň Region selling grain on the market, only one sold grain to neighboring Bavaria. For instance, in the early 1770s, only livestock were sold to Bavaria (Hochedlinger and Tantner, 2005). More significant was the grain trade with coachmen, especially in the northern half of Bohemia (Chalupa, 1969). They transported grain from the inland to the borderlands. This dynamic shifted in shortage years, like 1771, when
coachmen purchased grain directly from servile villages.

Grain demand and prices were also affected by the economic policies of the state. Lom (1929) attributed the stable price increase after 1740 CE to the wars of 1740–1748 and 1756–1763 (see Sect. 2.1) and to support for industrial development, which heightened the food needs of a non-self-sufficient population. Following the financial edict of March 1811, the state set official grain prices: 30 guldens of Vienna currency for wheat, 19 guldens for rye, 14 guldens for barley, and 9 guldens
for oats. However, in practice, prices were often up to 5 guldens higher. Despite a good harvest in 1811 (Svobodová, 1996), food prices in the following months were double (Petrtyl, 1958). After the currency stabilization in 1818, speculation opportunities were limited, and prices fell (Lom, 1929).

Grain prices were also influenced by the economic policies of local lordships. In cases where lordships, like some small estates around Sušice, did not own a brewery, they did not pressure their serfs and allowed them to freely sell their barley.
However, this freedom was complicated by the challenging conditions for barley cultivation in the area. For instance, in 1765, the barley price in Sušice was 25 kreutzers higher than in the Prague market (Lhoták, 2018). Furthermore, lordships converted monetary feudal annuities from cash to grain, a practice that became frequent on state and town domains, including Sušice, from the early 19th century (Lhoták, 2018). Conversely, serfs preferred to convert their in-kind feudal duties to cash, especially during times of high grain prices (Lom, 1929). However, the expansion of potato cultivation in the
submontane areas of southwestern Bohemia after the "hungry years" of 1770–1772 (*cf.* Brázdil et al., 2001) helped decrease the demand for bread grain.

To mitigate the negative impacts of harvest failure, the establishment of so-called contribution granaries was decreed in Bohemia in 1788. Each farmer was required to contribute annually one-third of the amount of four basic cereals, enough for three years of winter and spring grain sowing. In the event of a yield shortfall, a farmer could borrow grain, returning it later
with 12.5 % interest. While in Western Europe granaries were typically organized at the village level (e.g., Collet, 2010), in Bohemia, they were usually established in domain centers. However, their establishment often experienced significant delays (Černý, 1932).



### 5.3 Broader context

As indicated in Sect. 5.2, weather and climate are among the many factors influencing grain prices during the period
analyzed. In our study's region, we particularly discuss results in the context of several European studies that investigated the
relationship between weather, climate, and grain prices. Despite the underestimation of weather effects on grain harvests and
subsequent prices in Czech studies (*cf.* Borská-Urbánková, 1977), Brázdil and Durd'áková (2000) found that bad grain
harvests, connected with adverse weather patterns, accounted for two-thirds (67 %) of 61 selected years with extremely high
prices in the Moravian towns of Brno, Dačice, and Olomouc during the 16th–18th centuries. Socio-economic factors were
significant in over half of these years (57 %). However, in our study focusing on ten selected years with extremely high
prices in southwestern Bohemia (Sect. 4.2.2), only the years 1771–1772, 1802, and 1816–1817 were outside of war conflicts.
Our analysis of seasonal temperature and precipitation did not find statistically significant effects on grain prices in
southwestern Bohemia for 1725–1824 CE, except for MAM temperatures, which partially aligns with findings from other
European studies. For instance, Bauernfeind and Woitek (1999) noted the duration of the vegetation period (i.e., temperature
effects) as a crucial factor influencing annual grain prices in German cities like Nuremberg, Cologne, Augsburg, and Munich
during 1500–1599 CE. They also reported a particular positive impact of DJF precipitation and low SON temperatures on
prices. Camenisch (2015) found a significant influence of MAM and JJA temperatures on grain prices in the Burgundian
Low Countries during the 15th century: compound temperature effects explained up to 57 % of wheat price variability for
the Brussels series, and for rye, 51 % for Brussels and 62 % for Antwerp. Ljungqvist et al. (2022) in their extensive
statistical analysis of 56 grain price series in early modern Europe (excluding Czech Lands) for *c.* 1500–1800, found a
significant negative grain price–temperature relationship, with colder weather correlating with higher prices. The correlation
coefficient between the mean European grain prices and the previous year's June–August temperatures was –0.41 annually
and –0.63 on a decadal scale. However, precipitation and droughts showed only weak and spatially inconsistent signals in
European grain price series. These studies demonstrate that the effects of climate on grain harvests and prices can vary
significantly by region. This regional variation was further confirmed by Ljungqvist et al. (2023), who identified significant
signals in JJA soil moisture for Sweden, in DJF temperatures and precipitation for Switzerland, and in MAM and annual
mean temperatures for Spain in their climate–harvest yield associations.

In relation to the combined effect of temperature and precipitation as reflected in droughts, our study found statistically
significant positive correlation coefficients of grain prices with DJF, MAM, and JJA scPDSI (Sect. 4.2.1, Tables A1 and
A2). However, Esper et al. (2017) demonstrated for grain price series from 19 cities in central and southern Europe
(excluding Czech Lands) spanning the 14th to 18th centuries, that food shortages often coincided with regional JJA drought
anomalies. Notably, grain prices were exceptionally high during dry periods, even though the correlations of prices with tree
ring-based drought indices were very low, barely exceeding the value of −0.20. Similar low correlations were observed when
comparing series of three different drought indices for the Czech Lands with the grain price series of Dačice and Prague



(Brázdil et al., 2019). The only statistically significant correlation was found between the spring Standard Precipitation Index (SPI) and wheat prices in Dačice (1625–1802), with a correlation coefficient $r = -0.16$, $p < 0.05$.

## 6 Conclusion

The analysis of weather and climate effects on grain prices in southwestern Bohemia during 1725–1824 CE yielded the following key findings:

(i) Archival documentary data enabled the creation of grain price series (wheat, rye, barley, and oats) for Sušice in the period 1725–1824 CE. A few missing years were supplemented by corresponding Prague series, based on very high correlations between the two locations. Throughout the year, mean grain prices peaked in May–July before the harvest and decreased afterwards. Potential uncertainties stemming from the nature of the documentary data did not significantly affect the primary results of the statistical analyses or the identification of years with extremely high prices.

(ii) While weather extremes and climatic anomalies impacted grain harvests in Sušice during specific years, long-term climate variability only partly influenced fluctuations in grain prices. Relatively low Pearson correlation coefficients between Sušice grain prices and three Czech climate variables indicated a significantly stronger signal of cooler springs and wetter patterns from winter to summer in years with higher grain prices.

(iii) Years with exceptionally high grain prices in Sušice represented the combined effects of adverse weather patterns
contributing to poor grain harvests, along with the socio-economic and political conditions prevailing in Bohemia and Europe at the time. While the extreme years of 1746 and 1802–1806 coincided with wartime, the period of 1816–1817 followed the Tambora eruption in a post-war context, and the years 1771–1772 were not associated with such external influences.

(iv) Future analyses of climate/weather–harvest–price relationships should concentrate more on understanding local/regional
weather, climatic, and natural patterns, as well as territorially specific socio-economic and political phenomena, which may significantly impact such relationships. This emphasizes the importance of interdisciplinary research in examining these complex relationships at local and regional levels, as demonstrated in this study, and the need for close collaboration between climatologists and historians.



**Appendix A**

**Table A1. Pearson correlation coefficients between detrended grain-price series of Sušice and detrended climate variables (T – temperature, P – precipitation, scPDSI – self-calibrated Palmer Drought Severity Index) for winter (DJF), spring (MAM), and summer (JJA) during the period 1725–1824 CE. The statistical significance of correlation coefficients is indicated as follows: * p < 0.10, ** p < 0.05, *** p < 0.01.**

| Cereal | DJF | | | MAM | | | JJA | | |
|---|---|---|---|---|---|---|---|---|---|
| | T | P | scPDSI | T | P | scPDSI | T | P | scPDSI |
| wheat | 0.02 | 0.09 | 0.34*** | –0.25** | 0.10 | 0.31*** | –0.12 | 0.02 | 0.28*** |
| rye | 0.05 | 0.12 | 0.35*** | –0.26*** | 0.07 | 0.31*** | –0.12 | 0.04 | 0.29*** |
| barley | 0.06 | 0.09 | 0.31*** | –0.24** | 0.07 | 0.27*** | –0.06 | 0.01 | 0.25** |
| oats | –0.05 | 0.05 | 0.17* | –0.24** | –0.04 | 0.13 | 0.02 | –0.14 | 0.10 |


**Table A2. Pearson correlation coefficients between detrended grain-price series of Sušice and detrended temperature, precipitation, and self-calibrated Palmer Drought Severity Index (scPDSI) shifted by one year into the past for winter (DJF), spring (MAM), summer (JJA), and autumn (SON) during the period 1725–1824 CE. The statistical significance of correlation coefficients is indicated as follows: * p < 0.10, ** p < 0.05, *** p < 0.01.**

| Cereal | Temperature | | | | Precipitation | | | |
|---|---|---|---|---|---|---|---|---|
| | DJF | MAM | JJA | SON | DJF | MAM | JJA | SON |
| wheat | –0.02 | –0.16 | –0.15 | –0.06 | 0.18* | 0.15 | 0.22** | 0.08 |
| rye | –0.01 | –0.18* | –0.15 | –0.08 | 0.22** | 0.11 | 0.25** | 0.06 |
| barley | –0.04 | –0.19* | –0.14 | –0.06 | 0.20* | 0.09 | 0.21** | 0.07 |
| oats | –0.07 | –0.08 | 0.06 | –0.04 | 0.19* | 0.03 | 0.07 | 0.11 |


| Cereal | scPDSI | | | |
|---|---|---|---|---|
| | DJF | MAM | JJA | SON |
| wheat | 0.38*** | 0.38*** | 0.38*** | 0.35*** |
| rye | 0.39*** | 0.38*** | 0.39*** | 0.36*** |
| barley | 0.37*** | 0.35*** | 0.35*** | 0.32*** |
| oats | 0.23** | 0.21** | 0.18* | 0.18* |





**Table A3. Selection of years with extremely high prices in the detrended (high-pass filter) grain-price series of Sušice from 1725 to 1824 CE, based on the series' arithmetic mean increased by 0.5, 1.0, 1.5, and 2.0 multiples (M) of the standard deviation (SD).**
**Years above 0.5 SD are listed in the first line of the table. Years corresponding to higher multiples are denoted by the symbol "x".**

Wheat

| M | 1700+ | | | | | | | | | 1800+ | | | | | | | | |
|---|---|---|---|---|---|---|---|---|---|---|---|---|---|---|---|---|---|---|
|  | 43 | 45 | 46 | 47 | 58 | 59 | 71 | 72 | 90 | 02 | 03 | 04 | 05 | 06 | 12 | 13 | 16 | 17 |
| 1.0 |  |  | x |  |  |  | x | x |  | x | x | x | x | x |  |  | x | x |
| 1.5 |  |  |  |  |  |  | x | x |  |  |  |  | x | x |  |  | x | x |
| 2.0 |  |  |  |  |  |  | x |  |  |  |  |  | x | x |  |  | x | x |

Rye

| M | 1700+ | | | | | | | | | 1800+ | | | | | | | |
|---|---|---|---|---|---|---|---|---|---|---|---|---|---|---|---|---|---|
|  | 42 | 43 | 46 | 58 | 59 | 63 | 70 | 71 | 72 | 90 | 02 | 03 | 04 | 05 | 06 | 16 | 17 |
| 1.0 |  |  | x |  |  |  |  | x | x |  |  | x | x | x | x | x | x |
| 1.5 |  |  |  |  |  |  |  | x |  |  |  |  | x | x | x | x | x |
| 2.0 |  |  |  |  |  |  |  | x |  |  |  |  |  | x | x | x | x |

Barley

| M | 1700+ | | | | | | | | | 1800+ | | | | | | | |
|---|---|---|---|---|---|---|---|---|---|---|---|---|---|---|---|---|---|
|  | 43 | 46 | 47 | 58 | 59 | 63 | 70 | 71 | 72 | 90 | 02 | 03 | 04 | 05 | 06 | 16 | 17 |
| 1.0 |  | x |  |  |  |  |  | x | x |  | x | x | x | x | x | x | x |
| 1.5 |  | x |  |  |  |  |  | x | x |  |  |  | x | x | x | x | x |
| 2.0 |  |  |  |  |  |  |  | x |  |  |  |  |  | x | x |  | x |

Oats

| M | 1700+ | | | | | | | | | | 1800+ | | | | | | | | | | | | |
|---|---|---|---|---|---|---|---|---|---|---|---|---|---|---|---|---|---|---|---|---|---|---|---|
|  | 42 | 43 | 46 | 47 | 57 | 58 | 59 | 62 | 71 | 72 | 82 | 90 | 99 | 02 | 03 | 04 | 05 | 06 | 08 | 09 | 12 | 16 | 17 |
| 1.0 |  | x | x |  | x | x | x | x | x | x |  | x | x | x | x | x | x | x |  | x |  | x | x |
| 1.5 |  |  |  |  |  | x |  |  |  | x |  | x |  | x | x | x | x | x |  |  |  |  | x |
| 2.0 |  |  |  |  |  |  |  |  |  |  |  |  |  |  | x |  | x | x |  |  |  |  | x |

**Data availability.** The datasets and series used in this article are either publicly available (see citations of climate
reconstructions) or can be obtained by personal request.



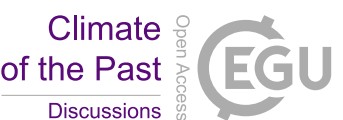

**Author contributions.** RB designed and wrote the paper with contributions from all co-authors. JL collected all grain-price data and contribute to paper writing by historical knowledge. KC did basic calculations concerning of grain-price and climate series and finalized all figures. PD did statistical analysis of grain-price and climate series. All authors have read and 590 commented on the latest version of the paper.

**Competing interests.** The contact author has declared that none of the authors has any competing interests.

**Special issue statement.** This article is not a part of the special issue. It is not associated with a conference.


**Acknowledgements.** Rudolf Brázdil was supported by the project "AdAgriF – Advanced methods of greenhouse gases emission reduction and sequestration in agriculture and forest landscape for climate change mitigation" (CZ.02.01.01/00/22_008/0004635). We are grateful to Lenka Hájková (Doksany) for supplying the data on grain harvests for the Mirotice and Nihošovice stations from the FENODATA database, to the Muzeum Šumavy in Sušice for providing Figure 600 1, and to Laughton Chandler (Charleston, SC) for his assistance with English style corrections.

**Financial support.** This research has been partly supported by the project "AdAgriF – Advanced methods of greenhouse gases emission reduction and sequestration in agriculture and forest landscape for climate change mitigation" (CZ.02.01.01/00/22_008/0004635).


**Archival sources**

AS1: Národní archiv, fond České gubernium-publicum, sign. 2/1, karton č. 24 and č. 152.

AS2: Národní archiv, fond Stavovský zemský výbor, inv. č. 12176, karton č. 166, sign. 1746/1/94; inv. č. 12274, karton č. 168, sign. 1746/2/98; inv. č. 12367, karton č. 169, sign. 1746/3/93; inv. č. 12451, karton č. 170, sign. 1746/4/84; inv. č. 610 12527, karton č. 171, sign. 1746/5/76; inv. č. 12596, karton č. 172, sign. 1746/6/67; inv. č. 12640, karton č. 173, sign. 1746/6/43; inv. č. 12693, karton č. 174, sign. 1746/8/45; inv. č. 12760, karton č. 176, sign. 1746/10/61; inv. č. 12821, karton č. 177, sign. 1746/11/52; inv. č. 12868, karton č. 179, sign. 1746/12/44; inv. č. 12882, karton č. 179, sign. 1746/12/59.

AS3: Státní oblastní archiv Litoměřice, fond Velkostatek Mimoň – Stráž pod Ralskem, inv. č. 72, Pamětní kniha učitele Františka Tomáše Spillara z Plzeňska z let 1771–1907 s přípiskem jeho pokračovatele k r. 1844.

AS4: Státní oblastní archiv Třeboň, fond Krajský úřad Prácheň, karton č. 78.

AS5: Státní oblastní archiv Třeboň, fond Krajský úřad Prácheň, karton č. 121.

AS6: Státní okresní archiv Klatovy, fond Archiv města Sušice, inv. č. 109, Manuál purkmistrovského úřadu 1698–1790.

AS7: Státní okresní archiv Klatovy, fond Archiv města Sušice, inv. č. 187, Kniha cen potravin v Sušici 1803–1827.

AS8: Státní okresní archiv Klatovy, fond Archiv města Sušice, inv. č. 945, karton č. 40.

AS9: Státní okresní archiv Klatovy, fond Farní škola Bukovník, inv. č. 25, sign. K 25, Pamětní kniha 1815–1869.





AS10: Státní okresní archiv Prachatice, fond Archiv obce Zálezly, inv. č. 1, kniha č. 1, Pamětní kniha 1791–1908.

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
