# Peer review of "Effects of weather and climate on fluctuations of grain prices in southwestern Bohemia, 1725–1824 CE"

_Climate of the Past, 2024_

## Author Comment (AC1)

**Responses to the review by Fredrik Charpentier Ljungqvist**

I can strongly recommend the article "Effects of weather and climate on fluctuations of grain prices in southwestern Bohemia, 1725–1824 CE" by Brázdil et al. for publication after only minor revision. It is clearly written, contains interesting results and are based, in part, on novel data. The structure of the article is clear. I have, however, recommendations for some additional references. Furthermore, I would like to see all material now placed in an Appendix included in the main article. It is only two tables and to include the in the main article would facilitate easier reading. My comments to individual things in the article as well as suggested additional references as are listed below.

RESPONSE: We would like to thank Fredrik Charpentier Ljungqvist for careful reviewing of our manuscript and minor critical comments which we are trying to respond below.

**Minor comments:**

Abstract, line 8 (and other places): Here the term "weather is used". How is this term used in relation to the term "climate". In my opinion, "climate" is a more suitable term in this article than "weather".

RESPONSE: Accepted and corrected as follows: "Grain prices in early modern Europe reflected the effects of weather and climate on crop yields and a complex array of societal and socio-economic factors."

Abstract, line 9: Would it be possible to be more precise than just writing "societal and socio-economic factors"?

RESPONSE: In the starting sentence of abstract we suppose that this general identification "societal and socio-economic factors" has its explanatory value and need not to be specified in detail here.

Abstract, line 13: The term "mean annual variation" is not very clear to me. Maybe it can be expressed in another way?

RESPONSE: Accepted and corrected as follows: "The mean highest prices during the year typically occurred from May to July …"

Line 21: I would even recommend the authors to state that grain was the single-most important food source and cite Scott (2017) to support this.

RESPONSE: Accepted, we added citation by Scott (2017). On the other hand, the expression "a vital food source" is strong enough to express importance of grain as a food source.

Line 22, after Person (1999), cite also Leijonhufvud (2001).

RESPONSE: Accepted and complemented.

Line 29: Cite also Skoglund (2024) here.

RESPONSE: Accepted and complemented.

Line 34: Maybe here also cite Ljungqvist et al. (2024).

RESPONSE: Accepted and complemented.

Line 39: It should be Edvinsson et al. (2009).

RESPONSE: Accepted and complemented.

Line 40: Cite also Huhtamaa et al. (2022).

RESPONSE: Accepted and complemented.

Line 104: This is a surprisingly low July temperature for Central Europe. Please discuss the effect of elevation here and state the elevation of the measurements.
RESPONSE: The first sentence in Sect. 2.1 reports position of Sušice "at the foothills of the Šumava Mountains … at an altitude of approximately 470 meters above sea level", i.e. from climatological point of view there is not "a surprisingly low July temperature for Central Europe" and it needs not to be discussed separately.

Line 115 (and elsewhere): Be clear why 1951–1980 is chosen as a reference period here.
RESPONSE: We used this period, because phenological data in CHMI database FENODATA were available only for this time. We believe that there is not necessary to explain it, because it is not further used in the paper. We used 1961–1990 reference, which is clearly explained in Sect. 3.2 by the following sentence: "Climate variables were expressed as series of deviations from the 1961–1990 reference period, which was preferred over the more recent 1991–2020 period, already significantly influenced by global warming (see Brázdil et al., 2022b)."

Lines 133–134: I think this can be part of the paragraph above.
RESPONSE: We believe that this sentence is on an appropriate place. The preceding part of this section concerns only of price data used for the creation of Sušice series, while on lines 133–134 we reported other grain price series from Prague.

Line 142: Please provide a reference to scPDSI as a metric.
RESPONSE: RESPONSE: Accepted and corrected as follows: "…the self-calibrated Palmer Drought Severity Index (scPDSI; Wells et al., 2004) was used."
Reference:
Wells, N., Goddard, S., and Hayes, M.: A self-calibrating Palmer Drought Severity Index, Journal of Climate, 17, 2335–2351, https://doi.org/10.1175/1520-0442(2004)017<2335:ASPDSI>2.0.CO;2, 2004.

Line 193: Please provide standard references to SEA.
RESPONSE: Accepted and corrected as follows: "SEA, originally proposed by Chree (1913) and later improved for the use (e.g., in paleoclimatology by Rao et al. 2019), is commonly employed to evaluate the mean response of climate variables to specific events …"
References:
Chree, C.: Some phenomena of sunspots and of terrestrial magnetism at Kew Observatory, Philosophical Transactions of the Royal Society of London, Series A, Containing Papers of a Mathematical or Physical Character, 212, 75–116, https://doi.org/10.1098/rsta.1913.0003, 1913.
Rao, M.P., Cook, E., Cook, B., Anchukaitis, K., D'Arrigo, R. Krusic, P., and LeGrande, A.: A double bootstrap approach to Superposed Epoch Analysis to evaluate response uncertainty. Dendrochronologia, 55, 119–124, https://doi.org/10.1016/j.dendro.2019.05.001, 2019.

Fig. 8: The panels in the figure are too small. I would suggest to redraw the figure so that "A" comes above "B" instead of having "A" and "B" side by side.
RESPONSE: Accepted and corrected as requested.

Line 500: I find "Western Europe" a somewhat problematic term here, especially as the reference Collet (2010) refers to Germany, as both Germany and Czechia can be said to be part of Central Europe.

RESPONSE: Accepted and corrected. The corresponding sentence was changed as follows: "For example, while in Prussia (Germany) granaries were typically organized at the village level (e.g., Collet, 2010), …"

Line 520: Ljungqvist et al. (2022) excluded Czechia because of the grain price series at hand were too short to fit the inclusion criteria. Maybe that should be stated.
RESPONSE: Accepted and corrected as follows: "(excluding Czech Lands due to the shorter grain price series available)".

Lines 525–527: Maybe state the direction of the correlations?
RESPONSE: Accepted and corrected as follows: "This regional variation was further confirmed by Ljungqvist et al. (2023), who identified in climate–harvest yield associations positive significant signals in JJA soil moisture for Sweden and warmer and drier DJF patterns for Switzerland, but negative in MAM and annual mean temperatures for Spain in their climate–harvest yield associations."

Line 542. Something is wrong, it seems, with the formulation "significantly stronger signal of cooler".
RESPONSE: Accepted. To clarify the related expression, the sentence was corrected as follows: "Pearson correlation coefficients between Sušice grain prices and three Czech climate variables indicated a statistically significant signal of cooler springs and wetter patterns from winter to summer in years with higher grain prices."

Page 23 and 24: Include in main article instead.
RESPONSE: Accepted, Tables A1–A3 were moved into the main text and Appendix A was cancelled.

Line 587 (and other places): Better to write "grain price" than "grain-price" with "-".
RESPONSE: We follow writing "grain-price" as done in English style corrections by our native speaker.

Line 602: Funding agency is lacking: only grant number and grant title provided.
RESPONSE: Accepted and complemented as: "This research has been partly supported by the Johannes Amos Comenius Programme and the Ministry of Education, Youth and Sports of the Czech Republic for the project "AdAgriF – Advanced methods of greenhouse gases emission reduction and sequestration in agriculture and forest landscape for climate change mitigation" (CZ.02.01.01/00/22_008/0004635)."

References:

Edvinsson, R., Leijonhufvud, L., and Söderberg, J.: Väder, skördar och priser i Sverige, in: Agrarhistoria på många sätt: 28 studier om människan och jorden. Festskrift till Janken Myrdal på hans 60-årsdag, edited by: Liljewall, B., Flygare, I. A., Lange, U., Ljunggren, L., and Söderberg, J., 115–136, The Royal Swedish Academy of Agriculture and Forestry, Stockholm, ISBN 978-91-85205-91-2, 2009.

Huhtamaa, H., Stoffel, M., and Corona, C.: Recession or resilience? Long-range socioeconomic consequences of the 17th century volcanic eruptions in northern Fennoscandia, Clim. Past, 18, 2077–2092, https://doi.org/10.5194/cp-18-2077-2022, 2022.

Leijonhufvud, L.: Grain Tithes and Manorial Yields in Early Modern Sweden: Trends and Patterns of Production and Productivity c. 1540–1680, PhD thesis, Swedish University of Agricultural Sciences, Ulltuna, ISBN 9157658293, 2001.

Ljungqvist, F. C., Seim, A., and Collet, D.: Famines in medieval and early modern Europe – Connecting climate and society, WIRES Clim. Change, 15, e859, https://doi.org/10.1002/wcc.859, 2024

Scott, J. C.: Against the Grain: A Deep History of the Earliest States, Yale University Press, New Haven, ISBN 9780300182910, 2017.

Skoglund, M. K.: The impact of drought on northern European pre-industrial agriculture, The Holocene, 34, 120–135, https://doi.org/10.1177/0959683623120, 2024.

---

## Author Comment (AC2)

The work of the manuscript CP-2024-2 is an excellent effort to integrate historical climate information with variables from social dimension.

The reviewer has no questions of detail about the work, methods and data, or the results.

RESPONSE: We would like to thank the anonymous referee #2 for evaluation of our paper and raising several critical comments, which we are trying to answer below.

The work could be published after addressing two minor issues:

**1) General question:**

The work focuses on cereal price fluctuations to identify climatic factors and their degree of incidence. Please can you explain if there are already works based on the cereal production variable in your study area. Can you comment/justify why you prefer use of market prices of different cereals to identify climatic factors incidence, instead use of cereal production information and statistics?

RESPONSE: We believe that the third paragraph of Introduction (lines 43–52) gives sufficient information about "works based on the cereal production variable in our study area". Concerning of the second part of the comment, grain prices are only one preserved information reflecting (although indirectly) grain production in long time series, standardly used for comparison with climate variables. Own farmers did not kept records about amount of production even in the second half of the 19th century. Also state administration did not collected information about agricultural production. It means that only some data about production coming from some farming bodies like nobility, church or towns, that have been owners of large fields, are available for the period analysed in our study. But such data can be only hardly used because they are rather epizodic and biased by other factors like changing areas of sowing or arable land (e.g., its renting or sale), variable proportions of individual cereals sown and systems of farming (e.g., three-field system or alternating).

Are there works already published about cereal production? Are these investigations ongoing? If not, can you give a brief summary of the availability of this type of information in documentary sources, and assess whether it would be possible to carry out a research such as the one you propose in this manuscript?

RESPONSE: There are no works about cereal production and such investigations are not provided in the Czech Republic. Grain production in the studied period was not primarily evaluated and was not long-term monitored. Reports of grain accounts and threshing registers giving information how much grain has been harvested and how much threshed have a torso-like character from the spatiotemporal point of view.

I consider this series of questions timely since to establish the incidence of the climate and its extreme meteorological events, its relationship with agricultural production data seems more consistent. Documentary sources can provide these values in time series, whether in statistics from government administrations, private documentation of farming families or, perhaps most valid in many regions, fiscal and ecclesiastical tithe records.

RESPONSE: As reported in two preceding points, sources continually recording production of agricultural crops on the state, land or regional levels do not exist in the Czech Lands for the period analysed (i.e., 1725–1824), i.e., no such long-term series are available. Concerning of tithe records, their majority is limited only to the establishing of norm (i.e. they determined how much individual peasants should pay, not how much they actually paid), but recording of really passed grain is rather rare and not available in any long-term time interval.

Summarising our responses to three preceding reviewer's points together, we understand well and appreciate his/her interest in obtaining other documentary sources giving information

about quantitative grain production in the Czech Lands. As follows from our responses, such data – if exists – suffer by great spatiotemporal inhomogeneity and it is very difficult to create any other utilisable time series. Because grain prices are only one source kept consistently for longer time, we used such newly created series of grain-prices for investigation of relationships with selected climate variables in our article. Being concentrated only on the use of this particular type of grain-related characteristics, we do not consider as necessary to discuss in the paper other potential sources related to any other characteristics of cereal production, which are clearly out of the scope of our manuscript.

**2) Specific question on methodological aspects:**
The co-authors demonstrate extensive knowledge and availability of previous materials, organized in databases. On the other hand, in the manuscript it is difficult to identify the information used to generate a series of data that are evidently reconstructed on temperature, precipitation and PSDI. They appear in figure 7, page 12.
RESPONSE: We reported corresponding series used in Fig. 7 in Section 2.3 Meteorological data. Here we give also corresponding citations of papers, in which reported series were characterised in a great detail, including basic data sources, i.e., readers interested in more details may look on these papers. But to follow this comment, we did corresponding changes in Section 3.3 as follows: "This includes the monthly temperature series for Central Europe, compiled from documentary-based temperature indices of Germany, Switzerland, and the Czech Lands in 1500–1854 and measured temperature series of 11 meteorological stations in Central Europe from 1760 onwards (Dobrovolný et al., 2010), which is fully representative for Bohemia (*cf*. Brázdil et al., 2022a). The seasonal precipitation series for the Czech Lands were derived from documentary-based precipitation indices for 1501–1854 and mean areal precipitation totals of the Czech Republic from 1804 onwards (Dobrovolný et al., 2015)."

The provenance of instrumental sources can be understood for the early instrumental period, approximately 1780-1825. On the other hand, the origin of the information on the same variables for the period 1720s-1780s or later is not identified. If indexes or previous materials are already available, it would be interesting for the public to briefly know their origin, characteristics and bibliographical references.
RESPONSE: We believe that our response to the preceding point can be understood also as our response to this particular point. To explain documentary sources that have been used for creation of corresponding temperature/precipitation index series is really out of the scope of our study, do not bringing any necessary information for fulling of the main aims of our paper.

---

## Author Comment (AC3)

Manuscript CP-2024-2 is excellent in terms of scientific contribution, scientific quality and presentation. The discussion and explanations are clear and of high quality, while providing a new, solidly substantiated contribution to the question of the relationship between weather, climate and grain prices (and also to the importance of grain prices as climate proxies). The social, political and economic dimensions of grain price formation are presented in detail. This work absolutely deserves to be published.

RESPONSE: We would like to thank the anonymous referee #3 for evaluation of our paper and raising several critical comments, which we are trying to answer below.

A few minor questions or comments are addressed to the authors, without seeking to embarrass them, insofar as the answers are only accessible if substantial sources are available to contextualise the events, which is not necessarily the case for the small town of Sušice at the time studied.

**1/ General questions :**

Insofar as the wars affecting the region are clearly indicated (lines 81-100), is it possible to know whether certain poor harvests - or even years with no information concerning Sušice (lines 400-403) - were the result of periods of occupation or troop movements?

RESPONSE: Because in southwestern Bohemia the import of inland grain by coachmen played only a negligible role, grain prices were primarily a result of situation at the local market and the market circuit. The effect of direct military presence in the town or in its immediate proximity on the grain prices could be considered (in the period analysed) particularly in 1742 and less in 1805, but it concerned only short-term military movements or passages. The price development could have been influenced more markedly by winter housing of army in war time (e.g., in Sušice in 1749) or during the long-term placing of troops (in Sušice during 1766–1779), because it increased demand for a few years.

Did troop movements, invasions (lines 460-464) or rumours of invasion cause harvests to be brought forward (or delayed), which could have an impact on the quantity and/or quality of harvests and therefore on price formation, a phenomenon already identified in relation to grape harvests (Labbé, T., Pfister, C., Brönnimann, S., Rousseau, D., Franke, J., and Bois, B.: The longest homogeneous series of grape harvest dates, Beaune 1354-2018, and its significance for the understanding of past and present climate, Clim. Past, 15, 1485-1501, https://doi.org/10.5194/cp-15-1485-2019, 2019.). Would it be possible, given the current state of the sources, to indicate (perhaps by a simple percentage) the number of harvests affected by this bias? Are these situations linked to high prices?

RESPONSE: The Sušice town was located in an outlying position outside of main land communications. It meant that passages of enemy armies were not comparable with levels achieved on important land communications (roads) in Bohemia going to Prague. The military situation influenced prices rather by supplying foodstuffs to military stores in regional central towns commended by the state. It was a usual practice for the whole time of war conflicts.

Conversely, in the same type of context and insofar as Figure 1 seems to show a town that was at least partly fortified, could the withdrawal of the population to the shelter of the town walls with its grain reserves artificially lower prices?

RESPONSE: The function of Sušice as a citadel was marginal in the period analysed. The last notes about defensive purpose of the town fortification are from 1620 and the Thirty Years War. But also in that time transfer of people into the fortified town was only short-term (hours or a few days), not causing any significant deviations in prices.

In times of peace, in the factors mentioned (lines 45-46) about years of exceptionally large harvests, might these not - counter-intuitively - cause prices to rise because of the extremely large workforce, means of transport and storage facilities that are mobilised during the harvests?

RESPONSE: The grain supplied to the Sušice town market (and generally to other town markets in Bohemia) stem from small producers (i.e., country peasants from their farms), that organised transport of grain on market themselves. Moreover, demand for grain in a given place did not changed importantly (the number of consumers was relatively stable), i.e., prices in case of good harvests decreased and consumer could choose from whom to buy grain.

**2/ Specific questions:**

Figure 1 could perhaps use a little commentary to go beyond the simple illustration and highlight what it shows in relation to the article: a small town on the banks of the Otava, in a peri-urban landscape of very heavily humanised hills, showing the crops mentioned in the article, with the Bohemian (?) forests remaining only on the summits.

RESPONSE: Accepted and changes as follows: "The town of Sušice on the Otava River from the southeast – drawing by Augustin Maštovský from 1850, based on an earlier sketch by his father, Josef Maštovský, from 1832. Unbuilt area of the town cadastre was used in agriculture for the grain production, which had not more than supplementary economic function for a part of Sušice inhabitants. Non-forested hills in the background indicate problems with wood that had to be transported to the town from the Šumava Mts."

Similarly, it might be worth highlighting the very good regional/local accounting suggested by Figure 4, with sources that have increasingly standardised administrative and fiscal norms? Could a reference to Figure 4 be placed on lines 148-149 (if that is what well what the figure illustrates)?

RESPONSE: Fig. 4 documents not centrally organised accountancy, but a working evidence originating during sending of official price reports to superior authorities, which had a standardized form, but were preserved only in torso-like form. Because reference to Fig. 4 is already on the beginning of Sect. 2.2, it seems not necessary to repeat it here (without direct relevance to the text).

Sušice was a redistribution market for 91 localities, which probably roughly make up its Hinterland (or even its Umland). Did the trade in grain to neighbouring regions capable of extending their economic and commercial influence to distant producing regions require the local authorities to close borders or introduce temporary legislation banning exports (especially in time of crisis)? (lines 474-480)

RESPONSE: The grain trade with areal or border overlap was minimal in the described region and grain was rather brought from the inland. If there appeared any orders permitting export, it followed from all-country decrees in the periods of wars or in the years of grain failure (e.g., 1771–1772).

One last question: are there any MAM or SON "killing frosts" that have impact on prices?

RESPONSE: You are right, that frosts can have some damaging effects on cereals. But in the analysed region we did not find any such indication in available documentary sources. From this reason we did not reported frosts in the first paragraph of Sect. 5.2 or in detail description of years with extremely high prices in Sect. 4.2.

---

## Author Comment (AC5)

It is very difficult to add comments to this paper after the previous reviewers!
This is a very interesting paper and should be published.
I have some very minor additions, which I don't think the other reviewers have bothered with, but since I have to write somehing, here are some very minor ideas and suggestions. Personally, I found table 2 and the examination of years with extremely high grain prices very interesting.
RESPONSE: We would like to thank the anonymous referee #4 for evaluation of our paper and raising several critical comments, which we are trying to answer below.

Some very minor language issues:
317: "... snow which was lying for four weeks that all grain already sown extinct..." something like? "...snow which was lying for four weeks and destroyed all grain already sown..."
RESPONSE: Accepted and changed as proposed.

318 "...i.e. sooner too wet and then too dry..." something like: "...i.e. first too wet and then too dry..."
RESPONSE: Accepted and changed as proposed.

References to the appendix (line 267 and particularly line 382) are not very clear. ("Table A1"). Eventually you come to these tables, but it would be easier if just "see appendix, table A1" - or even better, adopt Ljungqvist approach and place the Appendix in the text.
RESPONSE: Accepted. Tables from the Appendix were included in the text.

432 "raising grain prices to six guldens." This is an informative quote, but would be even more informative, if there was an addition to the "normal" price - or what price it was raised from.
RESPONSE: To present "normal" price is very difficult due to missing comparable values, because "raising grain price to six guldens" cites an author of the Klatovy chronicle. We may only mention situation from surrounding towns. The most extended bread grain (rye) at Sušice increased in the first half of 1741from 1.5 to 2 guldens. For 1742 rye prices are not available and before the harvest in 1743 it was 4.5 guldens. Prices in Klatovy could have been similar to the regional market in Písek, where rye prices increased from c. 4 guldens in January 1742 to 5.5 guldens before the harvest, while before the harvest in 1743 it was slightly less than 5 guldens. I.e., it would by speculative to add into chronicle citation any such value.

511. "... focusing on ten selected years /-/ only the years 1771-1772, 1802, and 1918-1817 were outside of war..." 5 out of 10 is not what I should describe as "only". Make it easy: delete "only".
RESPONSE: Accepted, "only" was deleted.